# Diagnostic Utility of Bronchoalveolar Lavage Flow Cytometric Leukocyte Profiling in Interstitial Lung Disease and Infection

**DOI:** 10.3390/biom15040597

**Published:** 2025-04-17

**Authors:** Erika M. Novoa-Bolivar, José A. Ros, Sonia Pérez-Fernández, José A. Campillo, Ruth López-Hernández, Rosana González-López, Inmaculada Ruiz-Lorente, Almudena Otálora-Alcaraz, Cristina Ortuño-Hernández, Lourdes Gimeno, Diana Ceballos-Francisco, Manuel Muro, Elena Solana-Martínez, Pablo Martínez-Camblor, Alfredo Minguela

**Affiliations:** 1Immunology Service, Virgen de la Arrixaca University Clinical Hospital (HCUVA), Biomedical Research Institute of Murcia Pascual Parrilla (IMIB), 30120 Murcia, Spain; e.m.novoab@hotmail.com (E.M.N.-B.); josea.campillo@carm.es (J.A.C.); ruth.lopez2@carm.es (R.L.-H.); rosana13a@hotmail.com (R.G.-L.); irl_98@hotmail.com (I.R.-L.); almudena.o.a@gmail.com (A.O.-A.); cristina.ortunoh@um.es (C.O.-H.); lourdes.gimeno@carm.es (L.G.); dceballosf@gmail.com (D.C.-F.); manuel.muro@carm.es (M.M.); 2Pneumology Service, Virgen de la Arrixaca University Clinical Hospital (HCUVA), Biomedical Research Institute of Murcia Pascual Parrilla (IMIB), 30120 Murcia, Spain; jarl77@yahoo.es (J.A.R.); elenasolana90@gmail.com (E.S.-M.); 3Department of Statistics and Operations Research and Mathematics Didactics, University of Oviedo, 33007 Asturias, Spain; perezsonia@uniovi.es; 4Department of Biomedical Data Science, Geisel School of Medicine at Dartmouth, 7 Lebanon Street, Suite 309, Hinman Box 7261, Hanover, NH 03755, USA; pablo.martinez-camblor@hitchcock.org

**Keywords:** interstitial lung disease, lung infection, flow cytometry, bronchoalveolar lavage, leukocyte profile, lung fibrosis

## Abstract

Interstitial lung diseases (ILD) represent a diverse group of disorders that primarily affect the pulmonary interstitium and, less commonly, involve the alveolar and vascular epithelium. Overlapping clinical, radiological and histopathological features make proper classification difficult, requiring multiple complementary methodologies, including flow cytometry of bronchoalveolar lavages (BAL). This retrospective study analyzed BAL flow cytometry data from 1074 real-life patients, quantifying alveolar macrophages, CD4/CD8 lymphocytes, neutrophils, eosinophils, and CD1a+ Langerhans cells, with the aim of evaluating its diagnostic utility in ILD and pulmonary infection. Clustering and logistic regression analyses identified seven distinct leukocyte profiles: lymphocytic (associated with hypersensitivity pneumonitis, cryptogenic organizing pneumonia, and lymphocytic interstitial pneumonia), sarcoidosis, macrophagic (including nonspecific interstitial pneumonia, desquamative interstitial pneumonitis, pneumoconiosis, and unclassifiable ILD), neutrophilic (including usual interstitial pneumonia, respiratory bronchiolitis ILD, and acute interstitial pneumonia), infectious diseases, eosinophilic ILD, and Langerhans cell histiocytosis. The estimated leukocyte profiles were associated with different overall survival (OS) outcomes. Neutrophilic profiles, both infectious and non-infectious, correlated with poorer OS, particularly in patients without pulmonary fibrosis. Furthermore, corticosteroids and other immunosuppressive therapies did not show significant OS differences across leukocyte profiles. Although the gold standard in BAL cytology continues to be cytopathology, these results support BAL flow cytometry as a rapid and reliable complementary tool to aid in the classification of interstitial lung diseases based on immune cell profiles, providing valuable predictive information and contributing to personalized therapeutic approaches.

## 1. Introduction

The term interstitial lung disease (ILD) refers to a broad and diverse group of disorders that affect the lung parenchyma, presenting with overlapping clinical, radiographic, and histopathologic features. These conditions primarily involve the pulmonary interstitium, but can also extend to the alveolar and vascular epithelium less frequently [1,2]. Since establishing specific diagnosis can be difficult in many cases, the 2013 classification of the American Thoracic Society/European Respiratory Society (ATS/ERS) [3] gave great importance to the discussion between clinicians, radiologists and pathologists in the so-called “Multidisciplinary Committees” [4]. ILD etiologies range from known causes such as connective tissue disorders, offending drugs and environmental triggers, to unclassifiable and idiopathic causes [3,5].

Bronchoalveolar lavage (BAL) is a safe technique that offers useful information in the diagnosis of ILD, providing cellular and microbiological evidence to ruled out inflammatory, infectious or neoplastic processes [3,6]. BAL, radiology and pulmonary function tests are included in the diagnostic algorithm of ILDs [3]. Although BAL usefulness is controversial, it should be performed in suspected ILDs, when no radiological pattern compatible with usual interstitial pneumonia (UIP) is observed on the high-resolution computed tomography (HRCT) imaging of the thorax [7].

Specific inflammatory leukocyte patterns in the BAL can help in the diagnosis of ILD [7,8], even though such patterns sometimes can be nonspecific [7]. Lymphocyte predominance is characteristic in hypersensitivity pneumonitis (HP) [9], cryptogenic organized pneumonias (COP), lymphoid interstitial pneumonia (LIP) and sarcoidosis. High CD4/CD8 lymphocyte ratio help to differentiate sarcoidosis from the other lymphocytic ILDs [10,11,12,13]. In eosinophilic ILDs it is characteristic of the predominance of eosinophils [14]. Neutrophils are predominant in UIP, acute interstitial pneumonia (AIP), respiratory bronchiolitis ILD (RB-ILD) and in pulmonary infections [7,15]. Finally, alveolar macrophages predominate in desquamative interstitial pneumonitis (DIP), nonspecific interstitial pneumonia (NSIP), pneumoconiosis, unclassifiable ILD (U-ILD) and also in the pulmonary Langerhans cell histiocytosis (PLCH), although in the latter, an increase in CD1a positive Lanhergans cell is usually detected. In fibrotic ILDs, BAL can help to distinguish fibrotic forms due to HP from idiopathic pulmonary fibrosis (IPF) [16,17,18]. In addition, subtype leukocyte counting may have prognostic value [19,20].

The aim of this retrospective study is to evaluate the diagnostic utility of BAL leukocyte profiling in a large series in real-world patients with ILD or pulmonary infection. Unlike in other series [7], leukocyte count was performed using multiparametric flow cytometry, a rapid and accurate technique. The study defines cellular patterns that help in the diagnosis of ILDs. The rapidity of flow cytometry in delivering results can contribute in the multidisciplinary discussions, avoiding in some cases the need for further invasive tests.

## 2. Materials and Methods

### 2.1. Patients and Samples

This retrospective observational study, conducted in real-world clinical settings, encompassed clinical, radiological, histopathological, and pulmonary function data from patients across 8 public hospitals in Spain’s Murcia Region. The study analyzed 1483 consecutive BAL samples analyzed by flow cytometry between 2000 and 2018, obtained following the American Thoracic Society’s clinical practice guidelines for BAL sampling [7]. Patients with lung or other cancers (211), asthma (86), chronic obstructive pulmonary disease (82), or tuberculosis (30) were excluded. The final analysis included 731 BAL samples from ILD patients and 231 from patients with pulmonary infections. A control group of 112 BAL samples was derived from patients initially suspected of lung disease (primarily due to radiological micronodules) but who showed no evidence of lung pathology after years of follow-up. Additionally, to provide a survival control representative of the general population, 243 patients with monoclonal gammopathy of undetermined significance (MGUS), free from pulmonary disease and adverse prognostic factors [21], were included for comparison of ILD patient outcomes.

Data collection for evolutionary analysis concluded on 1 April 2023. Evaluations included anamnesis, clinical examination, radiology (radiography and high-resolution computed tomography [HRCT]), BAL cytomorphology, microbiology, anatomopathological studies, and pulmonary function tests, conducted in alignment with standard protocols at participating hospitals. Pulmonary fibrosis was defined radiologically by the presence of reticular opacities, traction bronchiectasis, and honeycombing. ILD subtypes were classified following the ATS/ERS criteria [3], with idiopathic pulmonary fibrosis (IPF) categorized under the usual interstitial pneumonia (UIP) group (Table 1). Immunomodulatory therapies followed established guidelines [22]. Antifibrotic agents became accessible starting in 2014, but were limited by regional availability. Patients were stratified into three treatment groups based on electronic health records: (1) no systemic immunomodulatory therapy; (2) corticosteroid monotherapy; and (3) combined immunomodulatory regimens (e.g., rituximab, azathioprine, mycophenolate mofetil, cyclophosphamide, or tacrolimus) initiated after corticosteroids. The study received approval from the institutional review board (IRB-00005712). All participants singed a written informed consent in compliance with the Declaration of Helsinki.

### 2.2. BAL Leukocyte Profile Evaluated by Flow Cytometry

BAL samples collected from various hospitals in the Murcia Region were transported at 4–8 °C in insulated containers and processed immediately upon arrival at the flow cytometry laboratory. The volume and physical characteristics of each sample were documented. After centrifugation at 1800 rpm, the supernatant was removed, and the cell pellet was washed once with 15 mL of FACSFlow solution (Becton Dickinson [BD], San Jose, CA, USA). The pellet was then resuspended in 0.5 mL of ACSFlow, and a 50 μL aliquot immediately stained in TrueCount tubes (BD) with an antibody cocktail containing CD1a PE (HI149), CD3 BV510 (SK7), CD4 APC (SK3), CD8 PE-Cy7 (SK1), CD16 V450 (3G8), CD19 APC (SJ25C1), CD20 FITC (L27), CD45 APC-H7 (2D1), HLA-DR PerCp (L243) from BD, and CD66abce FITC (Kat4c) from Dako (Santa Clara, CA, USA). Samples were vortexed, incubated for 10 min at room temperature in darkness, and lysed with ammonium chloride solution (BD) for 7 min. Immediately, a minimum of 500,000 events were acquired using an 8-color FACSCanto-II flow cytometer (BD). Instrument calibration included daily adjustment of photomultiplier voltages with CS&T beads (BD) and fluorescence compensation adjustments every two months using FC beads (BD). Daily fine-tuning of compensation was performed using negative events as reference points for each fluorochrome [25].

To better reflect the physiological leukocyte populations in alveolar spaces, dead cell exclusion dyes were omitted. Both viable and non-viable cells (identified through reduced forward/side scatter [FSC/SSC]) were included in the analysis, provided they retained the expression of leukocyte-specific markers such as CD45. Data were analyzed using DiVA 9.0 ™ Software v9.0 (BD Biosciences, San Jose, CA, USA) following the gating strategy outlined in Figure 1.

### 2.3. Statistical Analysis

Data were organized using Excel v.17.0 (Microsoft Corporation, Redmond, WA, USA) and analyzed with R version 4.2.2 (R Foundation for Statistical Computing, Vienna, Austria) and SPSS 21.0 (IBM Corp., Armonk, NY, USA). The primary leukocyte subsets, treated as numerical variables, were summarized as mean ± standard error of the mean (SEM) and compared using ANOVA with Tukey’s HSD post hoc tests or Student’s *t*-test. Cluster analysis with three groups, based on k-means clustering of leukocyte subsets, and correspondence analysis were employed to examine associations between flow cytometry data and pathologies. Logistic regression models incorporating eight main leukocyte subsets were used to estimate pathology probabilities, with adjustments for data imbalance. The classification algorithm involved two steps when assigning patients to the group with the highest estimated probability. The diagnostic performance of the fitted logistic regression models and the proposed classification algorithm was evaluated by the receiver operating characteristic (ROC) curve and percentages of correct classification, respectively, resulting from leave-one-out cross-validation (LOOCV). Overall survival (OS) was defined as the time from the first BAL analysis to death or last follow-up for censored patients. Kaplan–Meier survival curves and Log-Rank tests were used for the survival analysis, with group outcomes expressed as the 75th percentile of OS (Q3). Statistical significance was set at *p* < 0.05.

## 3. Results

### 3.1. Clinical, Biological and Therapeutic Features of the Study Cohorts

The biological, clinical, and therapeutic characteristics of the study groups are summarized in Table 1. Age and sex were analogous across the study groups (controls, ILDs, and infectious pulmonary diseases). However, certain ILD subtypes were predominantly observed in men, including pneumoconiosis (96.3%), eosinophilic ILD (80.0%), IPF (71.7%), LIP (69.2%), and PLCH (66.7%). While age was generally similar across the groups, patients with PLCH were notably younger (36.7 years) compared to the average age of the other ILD pathologies (53.8 years). Active smoker status was present in around two thirds of patients (63.1% overall), more frequently among patients with PLCH (88.9%), pneumoconiosis (75%) and unclassifiable ILD (75.3%), less frequently in PH (40.5%). Appendix A shows the main and secondary radiologic imaging patterns and pulmonary locations in each pulmonary disease. In general patterns and locations are the expected for each pulmonary pathology [26].

### 3.2. Flow Cytometry of BAL Can Identify Different Leukocyte Profiles in ILDs and Infectious Diseases

First, the cellular content of the BAL was compared in samples from the local hospital that hosted the flow cytometry facilities of external hospitals (Figure 2A,B). It was found that with the flow cytometry method described in this manuscript, which included the dead cells, the leukocyte content of BALs from the local hospital, processed in 30 min, showed similar values to those transported refrigerated from external hospitals and processed in 60 to 90 min (Figure 2A). In contrast, smoking status induced slight but significant changes in the content of alveolar macrophages (70.2% vs. 61.6%, *p* < 0.001) and total lymphocytes (16.6% vs. 22.8%, *p* < 0.001) in smoker compared to non-smoker patients (Figure 2C). A comparative analysis revealed consistent leukocyte profiles in both local and external hospital cohorts when comparing smokers to nonsmokers (Figure 2D).

Figure 3A,B shows mean values of leukocyte subsets in BAL-control group and ILD or infectious pathologies, respectively. Combining cluster analysis (Figure 3C) and conventional statistics (Figure 3B) to flow cytometry data of BAL samples it was possible to differentiate 3 main clusters of leukocyte profiles: (1) lymphocytic ILDs: ILDs characterized by lymphocyte predominance including sarcoidosis, HP, LIP and COP; (2) macrophagic ILDs: ILDs with a predominance of alveolar macrophages and/or CD1a+ Languerhans cells including NSIP, DIP, pneumoconiosis, U-ILD and PLCH; and (3) neutrophilic ILDs: ILDs with a predominance of neutrophils and/or eosinophils including UIP, RB-ILD, AIP, infectious pulmonary diseases and eosinophilic ILD. Although a cluster analysis was not able to separate eosinophilic from neutrophilic or PLCH from macrophagic ILDs, counts of eosinophils and CD1a+ Languerhans cells were significatively higher in patients with eosinophilic ILD (30.1 ± 5.0% vs. 1.5 ± 0.2%, *p* < 0.001) and PLCH (3.5 ± 1.4 vs. 0.72 ± 0.2%, *p* < 0.01).

### 3.3. BAL Leukocyte Profiles to Orientate Lung Disease Diagnosis

To assess the usefulness of the BAL leukocyte analysis via flow cytometry in guiding the diagnosis of pulmonary pathology, a logistic regression analysis was performed with a two steps classification algorithm. In the first step, the estimated probability of belonging to one of the five major leukocyte profile groups (lymphocytic ILDs, macrophagic ILDs, neutrophilic ILDs, eosinophilic ILDs or PLCH) was calculated for each patient and individuals assigned to the group with the highest probability (Figure 4A). Performance evaluation of the logistic regression models (ROC curve for the estimated probabilities) was performed for each leukocyte profile group. High area under the curve -AUC- (0.87, 0.80, 0.83, 0.89 and 0.87), sensitivity (81.0%, 80.1%, 72.8%, 82.3% and 77.8%), specificity (79.6%, 65.3%, 82.8%, 96.1% and 82.3%) and successful classification (65.9%, 58.6%, 58.1%, 76.5% and 55.6%) was observed for lymphocytic, macrophagic, neutrophilic, eosinophilic or PLCH leukocyte profiles, respectively (Figure 4B).

In the second step, the estimated probability of belonging to sarcoidosis, HP and LIP/COP subgroups was calculated for those patients classified as “Lymphocytic-ILD” in the first step. Patients were assigned to the group with the highest estimated probability. The same process was applied to patients classified in the first step as “Neutrophilic ILDs”, which were further subdivided into infectious pulmonary disease or other neutrophilic-ILDs (Figure 5A). Performance evaluation of the logistic regression models (ROC curve for the estimated probabilities) was performed for each leukocyte profile group. High AUC (0.86, 0.65, 0.74, 0.80 and 0.69), sensitivity (82.7%, 59.5%, 78.0%, 68.9% and 88.2%) and specificity (77.5%, 72.3%, 68.9%, 78.5% and 41.2%) was observed for sarcoidosis, HP, LIP/COP, infectious disease and other neutrophilic leukocyte profiles, respectively. Successful classification of 56.1%, 27.1%, 32.1%, 65.8% and 17.8% was observed for each aforementioned leukocyte profile (Figure 5B).

The logistic regression formulas are accessible to pulmonologists around the world at https://bal-ildcalculator.imib.es/ (accessed on 21 February 2025), where they will be able to enter BAL leukocyte subset counts of individual patients to estimate the pathology most likely associated with this cellular profile. Examples of actual cases are illustrated in Appendix A.

Additionally, to evaluate the performance of the logistic regression models, a heatmap is presented with the estimated probabilities from all models for each individual (Figure 6).

### 3.4. Impact of BAL Leukocyte Profile in the Survival of ILD Patients

As expected, patients with normal leukocyte profiles (BAL-control) showed 75-pertentile OS (Q3), comparable to general population (10.6 ± 0.9 years vs. not-reached), but patients with ILD (5.2 ± 0.5 years) or infectious pulmonary disease (3.1 ± 0.6) showed shorter OS (*p* < 0.001) (Figure 7A). Neutrophilic (2.3 ± 0.9 years, *p* < 0.001) and infectious (2.2 ± 0.5 years, *p* < 0.001) BAL leukocyte profiles were associated with the shortest OS, followed by eosinophilic (5.1 ± 1.1 years), macrophagic (4.9 ± 0.7 years) and PLCH (5.8 ± 1.5 years). On the other hand, lymphocytic profiles showed the longest OS: sarcoidosis (8.5 ± 1.8 years), HP (7.2 ± 1.7 years) and LIP/COP (7.3 ± 1.3 years) (Figure 7B). Excluding infectious diseases, the impact of the neutrophilic profile, compared to the rest of BAL leukocyte profiles, was more evident in non-fibrotic ILDs (4.6 ± 1.2 vs. 13.1 ± 1.5 years, *p* < 0.001) than in fibrotic ILDs (1.6 ± 0.4 vs. 2.8 ± 0.5 years, *p* = 0.145) (Figure 7C). Although no significant differences were noted in patient survival based on the type of systemic immunosuppression administered (corticosteroids versus other immunosuppressive therapies), systemic corticosteroid tended to have shorter OS (3.8 ± 0.8 vs. 5.9 ± 1.3, *p* = 0.197) in macrophagic BAL leukocyte profiles (Figure 7D).

## 4. Discussion

Complexity of interstitial lung pathology requires diverse complementary diagnostic tools for the correct diagnosis of patients [3]. A proper anamnesis together with clinical and radiological data offer the first approximation, requiring BAL cytology and even pulmonary histology in pathologies with greater clinical and diagnostic overlap. Although invasive procedures should be reserved for more complex cases, BAL is performed with high safety and its adequate analysis can provide conclusive information [7]. The gold standard in BAL cytologic continues to be cytopathology [7]. However, flow cytometry can provide quicker and more objective results [27] and, with the appropriate design of antibody panels, accurately quantify the main leukocyte subsets. Several studies have described the usefulness of flow cytometry for the discrimination of sarcoidosis from other lymphocytic pathologies [10,11,12,13] or even to perform leukocyte subset counting in diverse ILDs [28,29,30]. However, flow cytometry is still far from being a well-known and accepted methodology among pneumologists. The findings of this study, conducted with real-life medicine patients, demonstrate that flow cytometry is a practical and accessible methodology which, when performed correctly, can yield reliable results even with samples that require transportation and processing at external facilities. The speed of this methodology allows rapid therapeutic decisions to be made in acute pathologies, and its objectivity and precision can support the diagnostic suspicions of generic or specific ILDs.

Our results show that the leukocyte cytopathology profiles described in the ATS guideline [7] to define lymphocytic, sarcoidosis, neutrophilic, macrophagic, eosinophilic or Languerhans-cell-rich ILDs, using single-cell population cut-off points, can also be determined by flow cytometry, but in this case using multiparametric estimations in which accurate quantification of all leukocyte subsets are taken into account. Both the unguided cluster analysis and logistic regression allowed for the identification of lymphocytic pathologies, sarcoidosis, HP, LIP and COP, among which sarcoidosis showed the highest sensitivity and specificity. Eosinophilic ILD and PLCH could also be identified with high sensitivity and specificity. In the same way, neutrophilic pathologies, UIP, RB-ILD and AIP, were clearly recognized, among which infectious pulmonary disease was identified with high sensitivity and specificity. Although RB-ILD is typically included among the pathologies with a predominance of alveolar macrophages, in our retrospective study this entity was included among entities with a predominance of neutrophils, which leads us to suspect that some patients with infectious processes could have been incorrectly classified as RB-ILD. Finally, macrophagic pathologies, pneumoconiosis, DIP, NSIP and U-ILD were recognized in the absence of other leukocyte profiles and a predominance of alveolar macrophages. BAL samples from patients without pulmonary diseases showed leukocyte values within the expected range [7,31]. Importantly, a high concordance was demonstrated when these flow cytometry profiles were contrasted with diagnoses performed on real-life medicine.

The logistic regression formulas used in this work are accessible to pneumologists around the world at https://bal-ildcalculator.imib.es/ (accessed on 21 February 2025), where they will be able to evaluate retrospectively or prospectively their own patients and contribute to validate the results described in this manuscript. Estimation of the most likely pulmonary pathology based on the complete leukocyte profile could in some cases help the multidisciplinary committee to avoid the need for biopsies. For example, differentiating between the most frequent fibrosing pathologies, IPF and PH, with relative accuracy.

The 2013 ATS/ERS classification noted that a significant proportion of ILD patients are challenging to classify, often due to the presence of mixed lung injury patterns [3]. Consequently, a single patient may exhibit multiple pathologic and/or radiologic patterns. These diverse patterns can be observed within a single biopsy or across biopsies from different sites (e.g., UIP in one lobe and NSIP in another) [32], or even pathologic and radiologic patterns may sometimes differ. In smokers, multiple radiologic and histologic features can coexist, including PLCH, RB-ILD, DIP, UIP or NSIP and emphysema. In fact, in our series, an overlap between leukocyte profiles was detected in different proportions among pathologies. Several causes could be behind these overlaps: (1) the presence of infections leading to increased counts of neutrophil in non-neutrophilic ILDs; (2) difficult access to the most affected lung area leading to unaltered or non-representative leukocyte profiles; (3) early stages of the disease not yet associated with significant alterations of the leukocyte profile; (4) leukocyte profiles are not always included in BAL cytology reports; (5) misclassification of patient pathology due to overlapping symptomatology and radiological patterns; or even (6) progression from non-fibrotic ILDs to UIP or NSIP, among others.

Nonetheless, although a correct diagnostic classification is of the utmost importance for the adequate clinical management of patients, our results indicate that the BAL leukocyte profile by itself is associated with differential overall survival of patients and, therefore, may contribute to therapeutic decisions. In fact, lymphocytic profiles, especially sarcoidosis, are associated with survival periods that, although shorter than those of the general population, are the longest of all pulmonary pathologies in our series. Macrophagic, eosinophilic and PLCH leukocyte profiles show intermediate life expectancy. However, neutrophilic profiles are associated with shorter survivals, especially in patients without pulmonary fibrosis. In patients with established fibrosis (including IPF and other fibrosing non-IPF ILDs) the impact of the leukocyte profile was less evident. While neutrophil counts demonstrated significant correlations with both developments of pulmonary fibrosis and reduced survival in our cohort [23,24], these prognostic associations did not compromise the diagnostic validity of BAL flow cytometric leukocyte profiling for disease classification. In addition, patient survival did not significantly differ based on the type of systemic immunosuppressant administered (corticosteroids versus other immunosuppressive therapies), systemic corticosteroid tended to have shorter OS in macrophagic BAL leukocyte profiles. Nonetheless, these findings should be validated in larger, independent studies.

## 5. Conclusions

Notwithstanding the inherent limitations of this retrospective analysis, including incomplete clinical data collection, restricted coverage of ILD subtypes and standard constraints of BAL cytological studies, the results obtained in the analysis of BAL leukocyte profiles in a broad representative sample of the most common ILDs, seem to indicate that flow cytometry could complement radiological, cytopathological and clinical studies and contribute to a better diagnostic classification of patients. Flow cytometry has proven to be one of the most versatile and useful tools for the monitoring of cytoreductive and immunomodulatory therapies [33], and therefore its application to evaluate treatment responses should also be explored in pulmonary pathologies, beyond its usefulness in diagnosis.

## Figures and Tables

**Figure 1 biomolecules-15-00597-f001:**
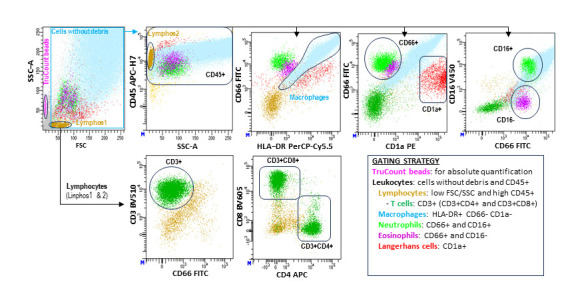
Flow cytometry analysis of leukocyte subsets in BAL samples. Leukocyte populations were characterized using a stepwise hierarchical and logical gating strategy (see figure). Absolute cell counts (cells/µL) were quantified with TruCount™ beads (BD) following the manufacturer’s protocol. This flow cytometry analysis method has been detailed in our group’s previous studies [24].

**Figure 2 biomolecules-15-00597-f002:**
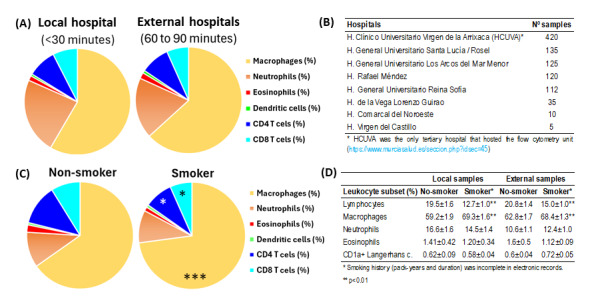
Impact of processing time and smoking status in BAL leukocyte content. (**A**) Comparison of leukocyte subsets in BAL samples between those processed within 30 min at the local hospital hosting the flow cytometry unit (*n* = 420) and those processed 60 to 90 min after extraction at external hospitals (*n* = 542). (**B**) Hospitals of origin for the samples. (**C**) Comparative results of leukocyte subsets in BAL samples from patients who did not smoke or who smoked. Pie charts represent the mean percentage values of leukocyte subsets. *, *p* < 0.05 and ***, *p* < 0.001 in the student *t*-test comparing non-smoker and smoker patients. (**D**) Comparative analysis of major leukocyte subsets in non-smoking versus smoking patients stratified by sample acquisition site (local vs. external hospitals). *, *p* < 0.01 in the student *t*-test comparing non-smoker and smoker patients.

**Figure 3 biomolecules-15-00597-f003:**
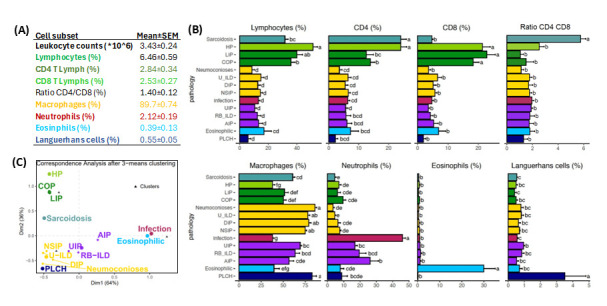
Groups of pathologies differentiated according to their BAL leukocyte profiles. (**A**) Mean ± SEM values of leukocyte subsets observed in BAL control group (*n* = 112). (**B**) Mean ± SEM of main leukocyte subsets in the BAL for ILD and infectious diseases distributed by groups according to the predominant leukocyte subset (green, lymphocytic; yellow, macrophagic; violet; neutrophilic; blue, eosinophilic; and dark-blue, PLCH). Letters a–g indicate different groups where patients would be allocated by their significant differences found by ANOVA and Tukey’s method in terms of the corresponding leukocyte subset. (**C**) Correspondence analysis biplot between ILD and clustering by k-means method performed on CD4 and CD8 lymphocytes, neutrophils, alveolar macrophages, eosinophils and CD1a+ Languerhans cells parameters. Distribution of different pathologies is shown according to similitude with the neighboring pathologies.

**Figure 4 biomolecules-15-00597-f004:**
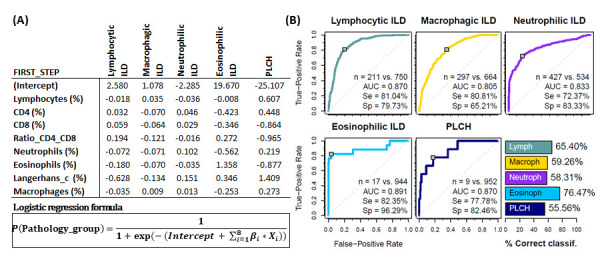
Groups of pathologies differentiated according to their BAL leukocyte profiles. First step logistic regression analysis. (**A**) Coefficients for logistic regression models *β*_*i* resulting from the first step (classification into one of the 5 major groups). Each column represents a different model for the corresponding pathology group. (**B**) Performance evaluation of the logistic regression models (ROC curve for the estimated probabilities) and the proposed classification algorithm (bar plot) via leave-one-out cross-validation. Number of patients, area under the curve (AUC) as well as sensitivity (Se) and specificity (Sp) are shown in each case.

**Figure 5 biomolecules-15-00597-f005:**
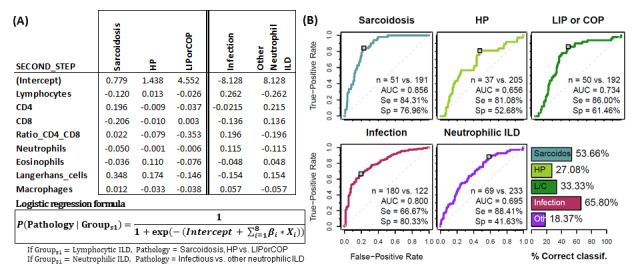
Groups of pathologies differentiated according to their BAL leukocyte profiles. Second step logistic regression analysis. (**A**) Coefficients for logistic regression models *β*_*i* resulting from the second step (first 3 columns if classification in first step is lymphocytic ILD, 2 last columns if classification in first step is neutrophilic-ILD). Each column represents a different model for the corresponding pathology group. (**B**) Performance evaluation of the logistic regression models (ROC curve for the estimated probabilities) and the proposed classification algorithm (bar plot) via leave-one-out cross-validation. Number of patients, area under the curve (AUC) as well as sensitivity (Se) and specificity (Sp) are shown in each case.

**Figure 6 biomolecules-15-00597-f006:**
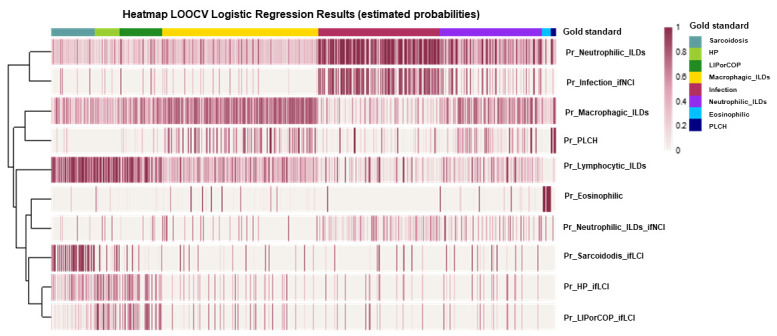
Performance evaluation of the logistic regression models through a heatmap of the estimated probabilities (by leave-one-out cross validation, LOOCV) from all models for each individual in the sample. Each vertical line (column) represents an individual, and each row indicates the response variable of the corresponding regression model.

**Figure 7 biomolecules-15-00597-f007:**
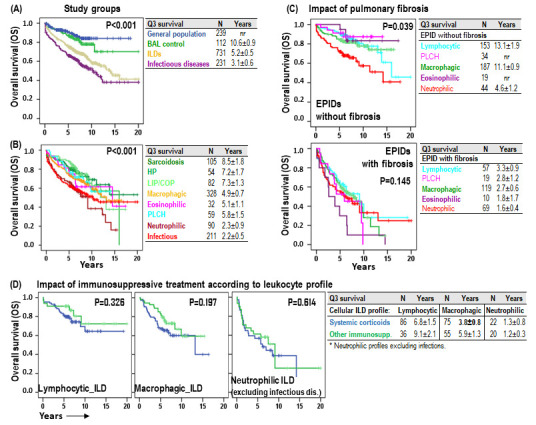
Impact of the leukocyte profile in the overall survival of ILD patients. (**A**) Kaplan–Meier survival curves and Log-Rank tests for overall survival (OS) according to the study group (general population, BAL–control, ILDs and infectious pulmonary diseases). (**B**) Kaplan–Meier and Log-Rank tests for OS according to the BAL leukocyte profile of each patient. (**C**) Kaplan–Meier and Log-Rank tests for OS according to the BAL leukocyte profile of each patient and to the presence of HRCT fibrosis patterns (fibrosis, reticular or honeycomb). (**D**) Kaplan–Meier survival curves and Log-Rank tests for OS according to the BAL leukocyte profile of each patient and to the type of systemic immunosuppressive therapy. Number of patients and 75-percentile OS (Q3) ± error is shown for each group.

**Table 1 biomolecules-15-00597-t001:** Biological, clinical and therapeutic characteristics of patient and control groups.

	Patients	Sex	Age	Fibrosis ^2^	Treatments (%) ^3^			
	(N) ^1^	(% man)	(Mean ± SD)	(%)	NT	IC	SC	Other
**Control** **groups**	**355**	**53.2%**	**61.2 ± 13.5**	-				
General population ^4^	243	53.1%	67.9 ± 12.3	-				
BAL-control ^5^	112	53.3%	53.5 ± 16.6	0%				
**Interstitial lung diseases (ILD)**	**731**	**58.1%**	**58.7 ± 16.4**	**38.2%**	**32**	**18**	**32**	**18**
Sarcoidosis	82	48.20%	55.6 ± 14.8	19.5%	34	13	38	15
Hypersensitivity pneumonitis (HP)	48	49.0%	50.4 ± 17.6	29.1%	13	28	46	13
Organized cryptogenic pneumonia (COP)	44	46.50%	62.3 ± 17.4	18.1%	18	9	52	21
Lymphocytic interstitial pneumonia (LIP)	37	69.20%	53.2 ± 15.9	8.1%	49	11	24	16
Usual interstitial pneumonia (UIP) ^6^	145	71.70%	65.8 ± 12.3	100%	25	21	35	19
Respiratory bronchiolitis ILD (RB-ILD)	25	40.0%	53.3 ± 22.8	8.0%	36	20	36	8
Desquamative interstitial pneumonitis (DIP)	35	52.8%	54.4 ± 20.8	5.7%	46	14	23	17
Nonspecific interstitial pneumonia (NSIP) ^7^	156	48.1%	59.3 ± 15.5	33.9%	31	11	28	30
Pneumoconiosis	27	96.3%	56.5 ± 16.0	18.5%	42	31	23	4
Pulmonary Langerhans cell histiocytosis (PLCH)	9	66.7%	36.7 ± 15.8	11.1%	67	11	11	11
Eosinophilic ILD	17	80.0%	48.4 ± 22.7	5.9%	0	18	76	6
Unclassifiable ILD (U-ILD)	80	60.0%	61.3 ± 12.4	25.0%	49	26	15	10
Acute interstitial pneumonia (AIP)	26	56.0%	57.5 ± 21.9	15.3%	29	35	24	12
**Pulmonary infectious diseases**	**231**	**59.0%**	**62.1 ± 15.6**	**15.1%**	**27**	**33**	**22**	**18**

^1^ The patient cohort in this study partially overlaps with those included in our group’s previous research [23,24]. ^2^ Radiological patterns associated with pulmonary fibrosis (fibrosis, reticular and honeycomb). ^3^ NT: no treatment; IC: inhaled corticoid; SC: systemic corticoid; other immunosuppressants: rituximab, azathioprine, mycophenolate mofetil, cyclophosphamide or tacrolimus. ^4^ Patients with monoclonal gammopathy with a good prognosis (without cytogenetic alterations or tumor circulating plasma cells in the peripheral blood) [21]. ^5^ BAL performed for etiological affiliation, but with no evident ILD pathology during follow-up. ^6^ Following ATS/ERS diagnostic criteria, idiopathic pulmonary fibrosis was included in the UIP group. ^7^ Patients with connective tissue disease ILD were included mostly in this group.

## Data Availability

The data that support the findings of this study are available from the corresponding author upon reasonable request.

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
