# Peer review of "Diagnostic Utility of Bronchoalveolar Lavage Flow Cytometric Leukocyte Profiling in Interstitial Lung Disease and Infection"

_biomolecules, 2025, doi:10.3390/biom15040597_

Round 1
Reviewer 1 Report
Comments and Suggestions for Authors
This is a clinically highly important paper describing data that suggest BAL flow cytometry as a useful tool to classify interstitial lung diseases. Results from this study were generated using high patient numbers.
The main question addressed by the research is the Value of BAL leukocyte profiling by flow cytometry in classifying interstitial lung diseases using high number of patients.
Interstitial lung diseases have traditionally be diagnosed using methods such as histological and radiological means. Such visual methods could be subjective and consistent assessment has been difficult. The quantitative unbiased method investigated in this study may provide improved assessment of this disease. Using high number of patients, the present study rigorously characterize the characteristics of this disease. Rigorous flow cytometry measurements provided critical information that has lead to authors’ conclusions.
Cited references are appropriate.
One question/suggestion is that the authors may address is if comorbid pulmonary hypertension and lung fibrosis influence authors results
Author Response
This is a clinically highly important paper describing data that suggest BAL flow cytometry as a useful tool to classify interstitial lung diseases. Results from this study were generated using high patient numbers.
The main question addressed by the research is the Value of BAL leukocyte profiling by flow cytometry in classifying interstitial lung diseases using high number of patients.
Interstitial lung diseases have traditionally be diagnosed using methods such as histological and radiological means. Such visual methods could be subjective and consistent assessment has been difficult. The quantitative unbiased method investigated in this study may provide improved assessment of this disease. Using high number of patients, the present study rigorously characterize the characteristics of this disease. Rigorous flow cytometry measurements provided critical information that has lead to authors’ conclusions.
Cited references are appropriate.
Coment-1: One question/suggestion is that the authors may address is if comorbid pulmonary hypertension and lung fibrosis influence authors results.
Response-1: Comorbid pulmonary hypertension was not systematically documented in all electronic registries, so we cannot provide detailed data on it. However, the impact of pulmonary fibrosis has been extensively analyzed in this series of patients by our group (references 23 and 24 of the manuscript). We have discussed this aspect in the last paragraph of the discussion section.
Reviewer 2 Report
Comments and Suggestions for Authors
In this manuscript, Novoa-Bolivar led a retrospective study that analyzed BAL flow cytometry data from 1074 real-life patients, quantifying several lung cells. They employed clustering and logistic regression analyses, identifying seven distinct leukocyte profiles.
The manuscript is well structured and professionally written.
Before considering a final opinion regarding its acceptance, I have some comments to be addressed by the authors.
In the abstract, include a brief line stating that the gold standard in BAL cytology continues to be cytopathology.
In the 2.1 Patients and Samples section, please include a table with data regarding each public hospital that refers patients and the number of samples for each. Additionally, include information about each center's level of care. I mean, are all centers level-3 hospitals?
In the 2.2 section, please specify the location for the "Laboratory" described in line 2 of the first paragraph.
In section 3.1, please add information about the frequency of exacerbations or if patients suffering from exacerbations were excluded.
The status of active smokers is really interesting. Please expand the given information, including years of smoking and pack years. Describe their influence on the results and discuss their importance in cell frequency.
Author Response
In this manuscript, Novoa-Bolivar led a retrospective study that analyzed BAL flow cytometry data from 1074 real-life patients, quantifying several lung cells. They employed clustering and logistic regression analyses, identifying seven distinct leukocyte profiles.
The manuscript is well structured and professionally written.
Before considering a final opinion regarding its acceptance, I have some comments to be addressed by the authors.
Coment-1: In the abstract, include a brief line stating that the gold standard in BAL cytology continues to be cytopathology.
Response-1: Done.
Coment-2: In the 2.1 Patients and Samples section, please include a table with data regarding each public hospital that refers patients and the number of samples for each. Additionally, include information about each center's level of care. I mean, are all centers level-3 hospitals?
In the 2.2 section, please specify the location for the "Laboratory" described in line 2 of the first paragraph.
Response-2: We have included all that information in Figure-2 since it nicely complements that figure. We believe that in this way it provides more information than an isolated table.
Coment-3: In section 3.1, please add information about the frequency of exacerbations or if patients suffering from exacerbations were excluded.
Response-3: Regrettably, this information was not consistently documented across all electronic records, and therefore, we are unable to provide detailed data on this matter.
Coment-4: The status of active smokers is really interesting. Please expand the given information, including years of smoking and pack years. Describe their influence on the results and discuss their importance in cell frequency.
Response-4: Again, information about number of packs and years of smoking was not consistently documented across all electronic records, and therefore, we are unable to provide detailed data on this matter, either. Extended information of cell content in smokers vs. non-smoker patients is now included in the figure-2, together with the local or external precedence of the sample and described accordingly in the manuscript.
Reviewer 3 Report
Comments and Suggestions for Authors
This retrospective study evaluated the diagnostic utility of bronchoalveolar lavage (BAL) leukocyte profiles analyzed by flow cytometry in 1074 patients with interstitial lung diseases (ILD) or lung infections. Seven distinct leukocyte profiles were identified, each associated with specific ILD subtypes (e.g., lymphocytic with hypersensitivity pneumonitis, neutrophilic with infections or fibrosis). The study demonstrated that flow cytometry provides rapid, reliable classification of ILDs, aiding multidisciplinary diagnosis and reducing the need for invasive procedures. Survival analysis revealed that neutrophilic profiles correlated with poorer outcomes, especially in non-fibrotic ILDs, while lymphocytic profiles had better prognoses. The findings support BAL flow cytometry as a valuable tool for ILD classification and prognostication, with logistic regression models available online for clinical use.
- The full terms rather than abbreviations should be used in the title.
- The specific aim of the study should be stated clearly in both the a abstract and the introduction.
- The conclusion should clearly refer to the limitations of this study and the limitations of BAL flow cytometry's diagnostic ability.
Author Response
This retrospective study evaluated the diagnostic utility of bronchoalveolar lavage (BAL) leukocyte profiles analyzed by flow cytometry in 1074 patients with interstitial lung diseases (ILD) or lung infections. Seven distinct leukocyte profiles were identified, each associated with specific ILD subtypes (e.g., lymphocytic with hypersensitivity pneumonitis, neutrophilic with infections or fibrosis). The study demonstrated that flow cytometry provides rapid, reliable classification of ILDs, aiding multidisciplinary diagnosis and reducing the need for invasive procedures. Survival analysis revealed that neutrophilic profiles correlated with poorer outcomes, especially in non-fibrotic ILDs, while lymphocytic profiles had better prognoses. The findings support BAL flow cytometry as a valuable tool for ILD classification and prognostication, with logistic regression models available online for clinical use.
Coment-1. The full terms rather than abbreviations should be used in the title.
Response-1: The title has been modified to try to make it somewhat shorter after including the full term instead of the abbreviations a: “Diagnostic utility of bronchoalveolar lavage flow cytometric leukocyte profiling in interstitial lung disease and infection”. Hope it is ok.
Coment-2. The specific aim of the study should be stated clearly in both the a abstract and the introduction.
Response-2: Done.
Coment-3. The conclusion should clearly refer to the limitations of this study and the limitations of BAL flow cytometry's diagnostic ability.
Response-3: Done.
Round 2
Reviewer 2 Report
Comments and Suggestions for Authors
Thanks for attending to my previous concerns.
Reviewer 3 Report
Comments and Suggestions for Authors
The required modifications have been performed.